# Understanding accelerators to improve SDG-related outcomes for adolescents—An investigation into the nature and quantum of additive effects of protective factors to guide policy making

Lorraine Sherr[1]⊙*, Katharina Haag[1]⊙, Mark Tomlinson[2,3]⊙, William E. Rudgard[4], Sarah Skeen[2], Franziska Meinck[5,6], Stefani M. Du Toit[2], Kathryn J. Steventon Roberts[1], Sarah L. Gordon[2], Chris Desmond[7], Lucie Cluver[8]

1 University College London, Institute for Global Health, London, United Kingdom, 2 Department of Global Health, Stellenbosch University, Institute for Life Course Health Research, Cape Town, South Africa, 3 School of Nursing and Midwifery, Queens University, Belfast, United Kingdom, 4 Department of Social Policy & Intervention, University of Oxford, Oxford, United Kingdom, 5 School of Social and Political Science, University of Edinburgh, Edinburgh, United Kingdom, 6 Faculty of Health Sciences, North-West University, Vanderbijlpark, South Africa, 7 University of KwaZulu-Natal, Durban, South Africa, 8 Department of Psychiatry and Mental Health, University of Cape Town, Cape Town, South Africa

⊙ These authors contributed equally to this work.
* l.sherr@ucl.ac.uk

**Data Availability Statement:** Data are formulated from a combination of the both the CCC study and

## Abstract

Recent evidence has shown support for the United Nations Development Programme (UNDP) accelerator concept, which highlights the need to identify interventions or programmatic areas that can affect multiple sustainable development goals (SDGs) at once to boost their achievement. These data have also clearly shown enhanced effects when interventions are used in combination, above and beyond the effect of single interventions. However, detailed knowledge is now required on optimum combinations and relative gain in order to derive policy guidance. Which accelerators work for which outcomes, what combinations are optimum, and how many combinations are needed to maximise effect? The current study utilised pooled data from the Young Carers ($n$ = 1402) and Child Community Care ($n$ = 446) studies. Data were collected at baseline ($n$ = 1848) and at a 1 to 1.5- year follow-up ($n$ = 1740) from children and young adolescents aged 9–13 years, living in South Africa. Measures in common between the two databases were used to generate five accelerators (caregiver praise, caregiver monitoring, food security, living in a safe community, and access to community-based organizations) and to investigate their additive effects on 14 SDG-related outcomes. Predicted probabilities and predicted probability differences were calculated for each SDG outcome under the presence of none to five accelerators to determine optimal combinations. Results show that various accelerator combinations are effective, though different combinations are needed for different outcomes. Some accelerators ramified across multiple outcomes. Overall, the presence of up to three accelerators was associated with marked improvements over multiple outcomes. The benefit of targeting

the YC study and ethically restricted from public sharing in accordance with special child protection and HIV protection provisions. Access enquiries should be addressed to the principal investigators for both studies (Lorraine Sherr and Mark Tomlinson for the CCC study and Lucie Cluver and Pamela Parath for the YC study). Here are the contact details for the authors administering our two databases: Lorraine Sherr- l.sherr@ucl.ac.uk Mark Tomlinson- markt@sun.ac.za Lucie Cluver - lucie.cluver@spi.ox.ac.uk Pamela Parath - pamela.parath@spi.ox.ac.uk.

**Funding:** The "UKRI GCRF Accelerating Achievement for Africa's Adolescents Hub" funded this work (grant provided through the Economic and Social Research Council, Grant number: ES/S008101/1). YC was funded by the UK Economic and Social Research Council and South African National Research Foundation (RES-062-23-2068), HEARD at the University of KwaZulu-Natal, the South African National Department of Social Development, the Claude Leon Foundation, the John Fell Fund, and the Nuffield Foundation (CPF/41513). Additional support was provided to LC by European Research Council (ERC) under the Europe Union's Seventh Framework Programme (FP7/2007- 2013)/ERC grant agreement n°313421 and the Philip Leverhulme Trust (PLP-2014-095). CCC was funded by Sweden/Norad through a nesting agreement with HelpAge. Partner organisations for CBO recruitment included World Vision, Bernard van Leer Foundation, Firelight Foundation, Save the Children, UNICEF, REPSSI, Help Age, Stop AIDS Now, AIDS Alliance, The Diana Memorial Fund, and Comic Relief. The Coalition for Children affected by AIDS provided support in the study initiation. UNICEF provided funding to partially support the creation of a combined data base. KJR is supported by a studentship from the Economic Social Research Council (ESRC), awarded through the UBEL-DTP. No funder, from either study, played a role in data collection, analysis, interpretation, writing the report, or the decision to submit the article for publication.

**Competing interests:** The author have declared that no competing interests exist.

access to additional accelerators, with additional costs, needs to be weighed against the relative gains to be achieved with high quality but focused interventions. In conclusion, the current data show the detailed impact of various protective factors and provides implementation guidance for policy makers in targeting and distributing interventions to maximise effect and expenditure. Future work should investigate multiplicative effects and synergistic interactions between accelerators.

## Introduction

Adolescents living in Sub-Saharan Africa are a particularly vulnerable group, affected by a range of adversities including poverty, violence exposure, high rates of HIV infection and poor access to services [1–3]. In order to meet the United Nations Development Programme (UNDP) Sustainable Developmental Goals (SDGs) for this group, there is a need to improve services and examine scale up of provisions. Current interventions are frequently delivered in a siloed manner [4], such as cash transfers to reduce poverty, or parenting interventions to reduce child violence experience. However, there is a growing body of evidence suggesting that combined interventions provide fundamental and sustained benefits, such as combinations including cash grants (e.g., cash plus care) [5,6] or food programmes together with parenting [7]. Based on such encouraging findings, recent work has aimed to identify potential accelerators for achieving SDG outcomes [8].

Accelerators are conceptualised as attainable actions (i.e., interventions, policies, practices) that have a simultaneous, cumulative effects across a range of outcomes [9]. Importantly, the UNDP SDG Accelerator and Bottleneck assessment distinguishes between accelerators, defined as life circumstances/ protective factors that directly influence SDG outcomes (e.g., food security or good parenting), and interventions, which can drive progress on access to these accelerators (e.g., feeding or parenting programmes) [10]. Evidence using observational data has identified a range of protective factors and their combinations (accelerator synergies) that bolster multiple outcomes for children and adolescents and can thus contribute to the accelerator model: safe schools [8], good parenting/parenting support [8,11], cash transfers [12], food security [11–13], living in a safe community [12,13], good mental health [14], education, ICT access [15], no survival work, small family size, food security, health extension [16], healthy caregivers [17] and access to community-based organisations [13]. There are now also programs emerging that are based on the idea of layered evidence-based services, such as the DREAMS programme, which aims to address the complex problem of high HIV incidences through a combined portfolio of provisions [18].

As the evidence base grows and shows solid, repeatable findings, policymakers are facing the challenge of how to adapt these findings to their respective contexts and how to plan comprehensive and synergistic services. They need to have clear, detailed guidance on how to choose between available programmes, how to optimise inputs and increasingly, in the context of COVID-19, on how to streamline services as resources shrink and demand increases. There is a strategic imperative to target multiple goals at the same time if the Sustainable Development Goals are to be achieved [19], while the realities of the COVID-19 pandemic have clearly indicated economic downturns and resource constraints in the face of increasing need [20–22]. A next step in understanding is to explore what the quantum of benefit is, i.e., whether there is an optimal number of combined services that achieve the best cost-benefit output, and to identify evidence that can guide decisions on which interventions to combine for which outcomes.

The current study aimed to explore the cumulative benefits of multiple accelerators on a range of SDG outcomes, and to understand the optimum number of combinations and the best clusters of combinations for strategic adoption and planning. It utilizes secondary observational data from two existing databases to gain initial evidence on what combinations of accelerators may be effective in improving specific SDG outcomes. Such evidence is vital for prioritising future research into interventions able to act on promising combinations of accelerators.

## Materials and methods

### Sample and procedure

The current study pooled data from two longitudinal studies, which were designed in close collaboration and utilized the same or similar measures of relevant constructs, to generate a sample of 1848 young people ($n$ = 1740 follow-up– 94.2% rate). Of these, 446 were drawn from the Child Community Care (CCC) study, and 1,402 from the Young Carers (YC) study. The CCC study investigated effects of attendance of community-based organisations (CBOs) in three countries (South Africa, Malawi, Zambia–only South African data is used in this paper) on child and adolescent outcomes, with high study enrolment (99.0%) at baseline and retention at a 12–18 month follow-up (86.0%). The YC study focused on the well-being of adolescents from disadvantaged backgrounds living in South Africa. Participants were drawn randomly from four census enumeration areas, with one child chosen at random from all visited households. Enrolment was 97.5% at baseline, and 96.8% were retained at the 1-year follow up. Adolescents in this study received no CBO support at either baseline or follow up. Detailed methods of recruitment for both studies are described elsewhere [23,24].

To pool both databases into one overall sample, only young people living in South Africa aged 9–13 years (overlapping age range between both studies) were selected. All data were obtained by trained data collectors in participant's language of choice and all participants and their caretakers provided written consent. Ethical approval for the YC study was obtained from the Universities of Oxford (SSD/CUREC2/11-40) and Cape Town (Ref: CSSR 389/2009), and the respective provincial Health and Education Departments. For the CCC study, ethical approval was obtained from University College London (1478/002) and Stellenbosch University (N10/04/112), as well as the funding agencies supporting the participating CBOs.

### Measures

**Accelerating protective factors.** The current analyses build on a previous manuscript [13], in which path analyses were used to investigate the effects of seven hypothesized protective factors on 14 SDG-related outcomes. All protective factors were hypothesized to simultaneously affect various SDG outcomes and thus to act as accelerators. Each factor had to be present at both baseline and follow-up to count as a potential accelerator, given evidence of the importance of sustained provision during child developmental progression [8]. Measures used and coding decisions for all protective factors and SDG outcomes can be found in the original paper [13]. The hypothesized protective factors comprised: 1) food security, coded as present if the child had not gone to bed hungry recently, 2) receipt of at least one of five government-provided cash grants over the past year in the household the child lived in (measured at follow up to cover the preceding year), 3) living in a safe community, indicated by children not witnessing or directly being exposed to community violence, 4) consistent access to healthcare when needed, 5) regular caregiver praise, 6) caregiver monitoring of child activities, and 7) access to CBOs, with the YC sub-sample specifically chosen to not have access to CBOs at any time-point, thus posing a comparison group.

**SDG-related outcomes.** Fourteen outcomes that aligned with five SDGs were retrospectively identified (for coding details: see [13]: S1 Table). This includes no symptoms major depression (MDD) and post-traumatic stress disorder (PTSD), no suicidality, as well as overall good mental health (combined score of all three previous measures), no peer problems, high prosocial behaviour (all SDG 3.4); no substance abuse (SDG 3.5); school enrolment, school attendance, being in the right grade for age, being able to concentrate at school (all SDGs 4.1/ 4.4); no sexual debut (given that the target population was relatively young) (SDG 5.6); no delinquent behaviours (due to their common link with aggression/violence grouped under SDG 16.1, "reduce violence everywhere"); and no exposure to physical and emotional abuse by the caregiver (SDG 16.2). Measures at follow-up were used as the main outcomes. We controlled for baseline score where possible to account for potential pre-existing differences between the YC and CCC samples. Exceptions were the peer problems and prosocial behaviour subscales and the sexual debut variable, for which measures were only available at follow-up.

**Covariates.** We included seven covariates in our analyses that were measured at baseline: child age, child sex, whether the family lived in formal versus informal housing, maternal/ paternal death, caregiver HIV-status and child caretaking responsibilities for other children or adults.

## Analysis

Descriptive analyses and analyses of those retained versus lost to follow-up were conducted using $\chi^2$ tests and two-tailed t-tests as appropriate. As described above, the current analyses build on previous work using path analyses to investigate the effect of seven hypothesized protective factors on 14 SDG-related indicators (standardized approach developed by Rudgard and colleagues 2020; code: https://osf.io/n6jy7/?view_only= 17f148085fde4b3fb645106c6c6e418b). After the absence of multi-collinearity between accelerators was established, separate multivariable logistic regressions were conducted, with each outcome being simultaneously regressed on all protective factors and covariates. The aim was to identify "accelerators", defined as protective factors that were related to three or more SDG outcomes after correcting for multiple testing via Benjamini Hochberg corrections (false discovery rate: 0.1) [25]. Accelerators affecting three or more outcomes identified in these analyses were: *caregiver praise*, *caregiver monitoring*, *living in a safe community*, *CBO access* and *food security*. The current paper presents a more in-depth follow-up analysis, focusing on what combinations of these five accelerators affected specific outcomes, and on identifying the quantum of benefit. For this purpose, adjusted probabilities and adjusted probability differences were calculated using the "margins" command in Stata, based on the original model (including covariates). We compared adjusted probabilities for the hypothesized presence of no accelerator, single accelerators and all possible combinations of accelerators to determine optimal combinations. We also calculated probability differences and associated confidence intervals to establish whether there were significant increases in predicted probabilities through the addition of additional accelerators. For this, we focused on the most effective accelerator combinations (i.e., the single most effective accelerator, combinations of the two to four most effective accelerators, all five accelerators). All analyses were conducted in Stata SE v.16.

## Results

### Descriptive statistics

Overall, $n$ = 108 participants (4.5%; $n$ = 63 from CCC, $n$ = 45 from YC) were lost to follow-up. This group was on average younger at baseline ($M$ = 11.23, $SD$ = 1.25, versus $M$ = 11.47,

**Table 1. Access rates to the hypothesized protective factors, SDG-related outcomes and socio-demographic covariates across baseline and follow-up.**

| | Baseline (N = 1848) | | | Follow-Up (N = 1740) |
|---|---|---|---|---|
| | Retained | Not retained | p-value | |
| **Hypothesized Protective Factors** | | | | |
| Caregiver Monitoring | 1011 (58.6%) | 71 (65.7%) | .118 | 1022 (58.9%) |
| Caregiver Praise | 1379 (79.3%) | 85 (78.7%) | .882 | 1180 (68.1%) |
| Safe Community | 1039 (59.8%) | 73 (67.6%) | .106 | 1050 (60.8%) |
| Healthcare Access | 1533 (88.1%) | 106 (98.2%) | .001* | 1523 (87.8%) |
| Any grant (FU only) | - | - | - | 1410 (81.1%) |
| Food security | 1446 (83.1%) | 88 (81.5%) | .663 | 1438 (83.0%) |
| CBO Access | 446 (24.1%) | 63 (58.3%) | < .001* | 383 (22.0%) |
| **SDG-Outcomes** | | | | |
| 3.4 No Depression | 1483 (85.2%) | 89 (82.4%) | .425 | 1488 (85.9%) |
| 3.4 No Suicidal Ideation | 1672 (96.1%) | 105 (97.2%) | .553 | 1665 (96.1%) |
| 3.4 No PTSD | 1635 (94.2%) | 104 (96.3%) | .358 | 1627 (93.9%) |
| 3.4 No Mental Health Problems | 1394 (80.3%) | 84 (77.8%) | .524 | 1379 (80.0%) |
| 3.4 No Peer Problems | - | - | - | 568 (32.9%) |
| 3.4 Prosocial Behaviour | - | - | - | 867 (50.1%) |
| 3.5 No Substance Abuse | 1382 (79.5%) | 91 (84.3%) | .230 | 1637 (94.5%) |
| 4.1/4.4 School enrolment | 1729 (99.4%) | 107 (99.1%) | .712 | 1697 (98.3%) |
| 4.1/4.4 School attendance | 1418 (82.4%) | 92 (86.0%) | .336 | 1663 (97.2%) |
| 4.1 / 4.4 Right Grade for Age | 999 (58.7%) | 59 (55.7%) | .538 | 957 (56.0%) |
| 4.1/ 4.4 Ability to Concentrate | 1374 (80.0%) | 80 (75.5%) | .263 | 1473 (84.8%) |
| 5.6 No Early Sexual Debut | - | - | - | 1438 (94.7%) |
| 16.1 No Violence Perpetration | 926 (53.4%) | 59 (54.6%) | .799 | 856 (49.3%) |
| 16.2 No Caregiver Abuse | 754 (43.4%) | 42 (38.9%) | .360 | 851 (48.9%) |
| **Sociodemographic Characteristics** | | | | |
| Child Sex (female) | 956 (54.9%) | 55 (50.9%) | .416 | 956 (54.9%) |
| Child Age (M, SD) | 11.47 (1.19) | 11.23 (1.25) | .046* | 12.71 (1.32) |
| Caregiver HIV Positive | 360 (20.7%) | 19 (17.6%) | .439 | 219 (12.6%)[+] |
| Informal Housing | 453 (26.0%) | 32 (29.6%) | .410 | 304 (17.5%) |
| Parental Death | 520 (30.0%) | 44 (41.1%) | .016* | 539 (31.2%) |
| Child Caretaker of Adults in HH | 502 (28.9%) | 35 (32.4%) | .436 | 445 (25.8%) |
| Child Caretaker of Younger Children | 438 (25.2%) | 39 (36.1%) | .012* | 474 (27.4%) |

*Notes.* Access to grants, sexual debut, prosocial score, and peer problems were assessed at follow-up only. Varying values due to missing data. HH = household, [+] drop in numbers may be due to changes in caregiver between baseline (BL) and follow-up (FU) (n = 401, 23.0%).

SD = 1.19, p = .046), more likely to have had a parent die (41.1% versus 30.0%, p = .016), and to be a caregiver of younger children (36.1% versus 25.2%, p = .012), but showed no differences on other sociodemographic variables. The retained sample was on average 11.47 (SD = 1.19) years old at baseline and 12.71 (SD = 1.32) years old at follow-up; 54.9% were female. Table 1 provides further demographic details. Only five of the investigated seven protective factors were related to three or more SDG outcomes in our original analyses (for details, see 13), and thus met our definition for being an accelerator. The following analyses therefore only focus on these five factors. Access rates to these accelerators across both timepoints lay at 22.0% (n = 383) for being enrolled in a CBO, 38.2% (n = 663) for caregiver monitoring, 53.9% (n = 930) for living in a safe community, 56.1% for caregiver praise (n = 971), 71.9% for food security (n = 1246), 77.9% (n = 1351) for healthcare access, and 81.1% for

cash grants. $n$ = 93 (5.4%) of the participants had access to none of the accelerators, $n$ = 313 (18.2%) to one, $n$ = 488 (28.4%) to two, $n$ = 482 (28.0%) to three, $n$ = 305 (17.7%) to four and $n$ = 40 (2.3%) to all five accelerators. Detailed data on how many participants had access to each accelerator combination (e.g., only food security but no other accelerators, or all five accelerators) are provided in S1 Table. Data in our initial paper [13] furthermore indicated that CBO attendance was associated positively with being food secure, receiving cash grants and healthcare access, and negatively with living in a safe community, and caregiver praise, though at small to medium effect sizes.

## Additive effects of different accelerator combinations

S2 Table presents a full table of the predicted probabilities and probability differences for achievement of the different SDG-outcomes based on our original model for the presence of i) no accelerators, ii) single accelerators and iii) various combinations of the accelerators. In the following, we will summarize the combinations of single, two, three, four and five accelerators that led to the strongest additive changes in predicted probabilities for each SDG-related outcome.

**Mental health.** The adjusted probability of experiencing no depression if no accelerator was present lay at 65.50%, which was raised to 97.94% if all five accelerators were present (see Fig 1). The strongest change by a single accelerator was made through CBO access (probability difference (PD): + 19.40 percentage points (pp)), which was raised to +27.62 pp if the child also lived in a safe community, leading to estimated rates of 93.12% experiencing no depression. For combinations of three and four accelerators, no strong additional changes were found. The adjusted probability for no suicidal ideation without any accelerators was 89.04%, which was raised to 99.36% when all five accelerators were present. Caregiver praise was the single accelerator leading to the largest changes (PD: + 6.24 pp). Combined with CBO access, this rose to +8.87 pp, equalling 97.91% with no suicidal ideation. After this, only small additional improvements were gained through adding food security (PD: +9.59 pp) and living in a safe community (+10.07 pp).

The adjusted probability for no PTSD without any accelerators lay at +91.45%. There was no evidence that this was significantly raised by the presence of all five accelerators (PD: +4.00 pp; confidence interval includes zero), due to negative associations of the outcome with caregiver monitoring (-6.23 pp). Caregiver praise (PD: + 4.30 pp) and the latter combined with living in a safe community (+5.72 pp; 97.17% with no PTSD) led to small changes in predicted probabilities for the outcome but including any additional accelerators did not yield strong increases. The adjusted probability for good mental health, a combination score of the previous three outcomes, improved from 59.07% for no accelerator being present to 95.11% with all five accelerators being present. In terms of single accelerators, living in a safe community (PD: +16.30 pp) and CBO access (PD: +18.59 pp) had similarly large effects, leading to a combined change of +29.07 pp (good mental health: 88.14%). This was raised to +33.89 pp by adding caregiver praise; with additional increases through CBO access (PD: +35.81 pp) and all five accelerators (+36.04 pp) being only marginal.

**Behavioural outcomes.** The adjusted probability for experiencing no peer problems lay at 10.39%, which was raised to 76.93% when all five accelerators were present. Increases were relatively equal, with the strongest changes found for CBO access as a single accelerator (PD: +22.94 pp). Adding caregiver monitoring (PD: + 34.44 pp), a three-accelerator-combination including living in safe communities (PD: + 50.04 pp) and a four-accelerator-combination including caregiver praise (+59.07 pp) improved adjusted probabilities even further (see Fig 2).

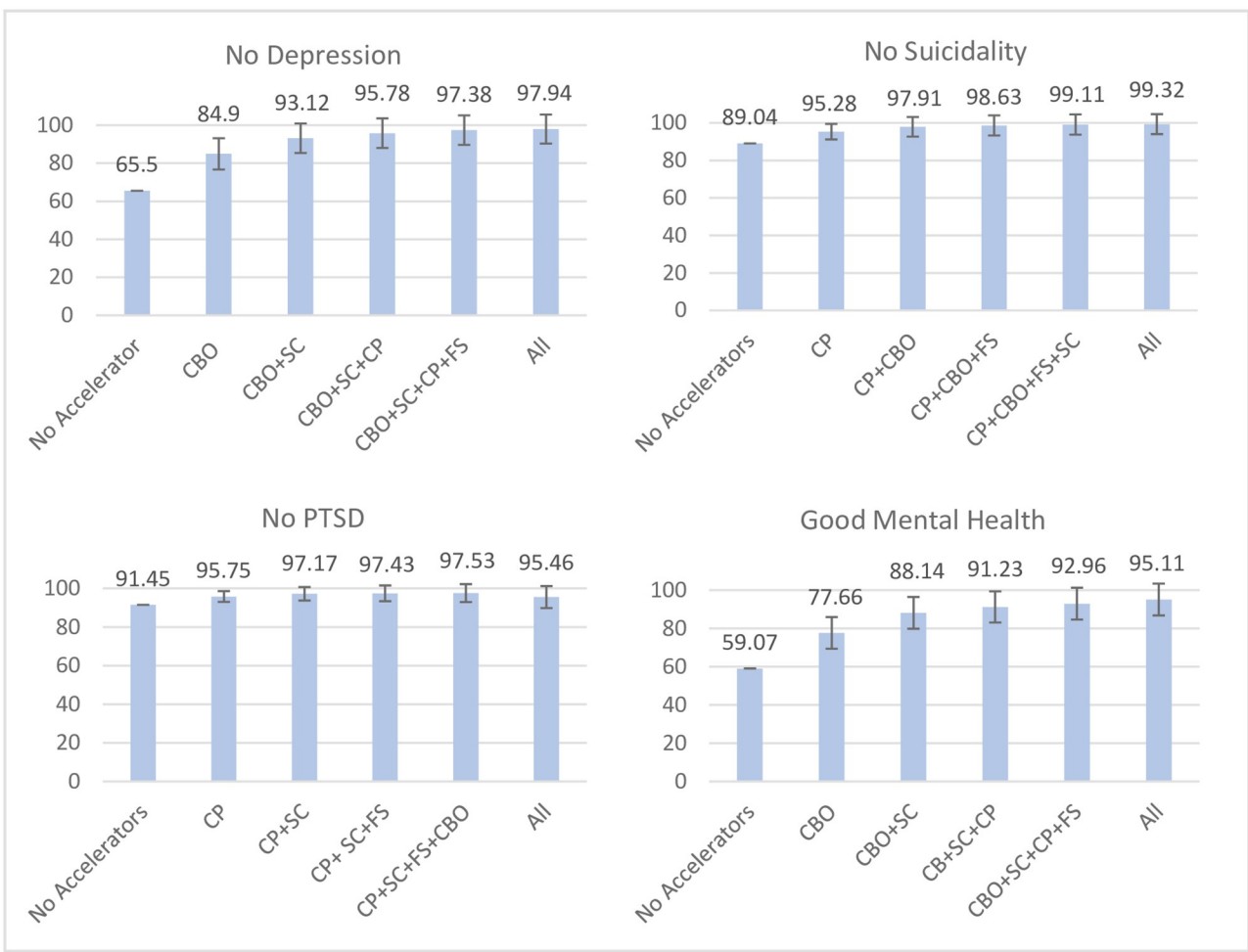

**Fig 1. Accelerator combinations leading to the strongest changes in adjusted probabilities for each mental health outcome.** Figure scales adjusted based on range of outcome values. Presence of all conditions based on symptom not clinical cut-offs (13). Good mental health: composite measure of the other three conditions. CBO = Access to community-based organisations, SC = Living in a Safe Community, CP = Caregiver Praise, FS = Food security, CM = Caregiver Monitoring. 95% confidence intervals included.

The adjusted probabilities for showing prosocial behaviour without any accelerators being present lay at 29.85%, which rose to 82.27% when all five accelerators were present. CBO access led to the strongest increase (+31.46 pp). In combination with caregiver praise, change lay at +41.20%, and when young people were additionally living in a safe community, at + 46.28 pp (prosocial behaviour: 76.13%). Adding food security (PD: +49.59 pp) and having access to all five accelerators (+52.43 pp) only led to small additional increases.

For no substance abuse, the adjusted probabilities lay at 92.86% for no accelerators being present, which was improved to 98.23% (PD: + 5.37 pp) when all accelerators but food security (negative association at -3.00 pp) were present (adjusted probability for all 5 accelerators: 97.40%). Single accelerator changes were largest for caregiver praise (PD: + 3.28 pp; rates of no substance use: 96.14%), with small additional changes evoked by adding CBO access (PD: +4.86 pp) and living in a safe community (PD: +5.76 pp). Adding more than three accelerators had negative effects, with caregiver monitoring on its own also being negatively associated (-1.83 pp).

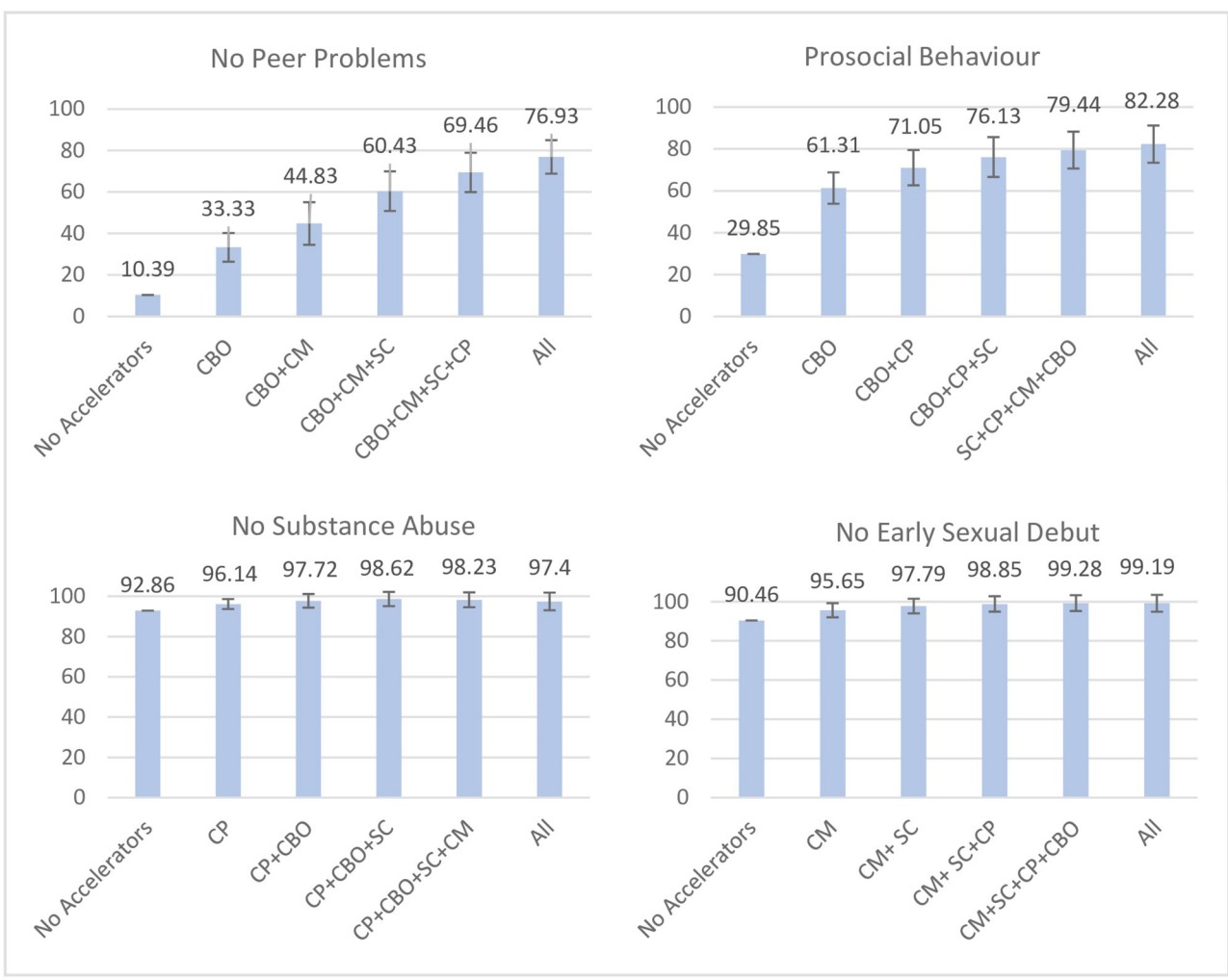

**Fig 2. Accelerator combinations leading to the strongest changes in adjusted probabilities for each behavioural outcome.** Figure scales adjusted based on range of outcome values. CBO = Access to community-based organisations, SC = Living in a Safe Community, CP = Caregiver Praise, FS = Food security, CM = Caregiver Monitoring. 95% confidence intervals included.

For <u>no early sexual debut,</u> predicted probabilities without any accelerators being present lay at 90.46%, which improved to 99.19% when all five accelerators were present. Largest changes were gained for caregiver monitoring as a single accelerator (PD: +5.19 pp), which increased to +7.33 pp when adding living in a safe community, and +8.39% when in addition caregiver praise was experienced (no early sexual debut: 98.85%). Additional increases through adding CBO access (PD: +8.82 pp) and all five accelerators (+8.73 pp) were marginal.

**School outcomes.** Rates of <u>school enrolment</u> were very high (98.3%) in this sample, leading to the outcome being dropped from path analyses, as it prevented model fit due to sparse cells. Rates of <u>school attendance</u> (i.e., not missing school) under the presence of no accelerators lay at 97.57%, which was not substantially improved by the presence of any accelerators (all 5 accelerators present: 96.21%, due to negative associations with food security (-.54 pp) and CBO access (-2.73 pp); see Fig 3). Without any accelerators being present, 54.34% of children were in the <u>right grade for their age;</u> this remained at 53.50% when all five accelerators were present due to negative associations of the outcome with CBO access (-3.80 pp), caregiver

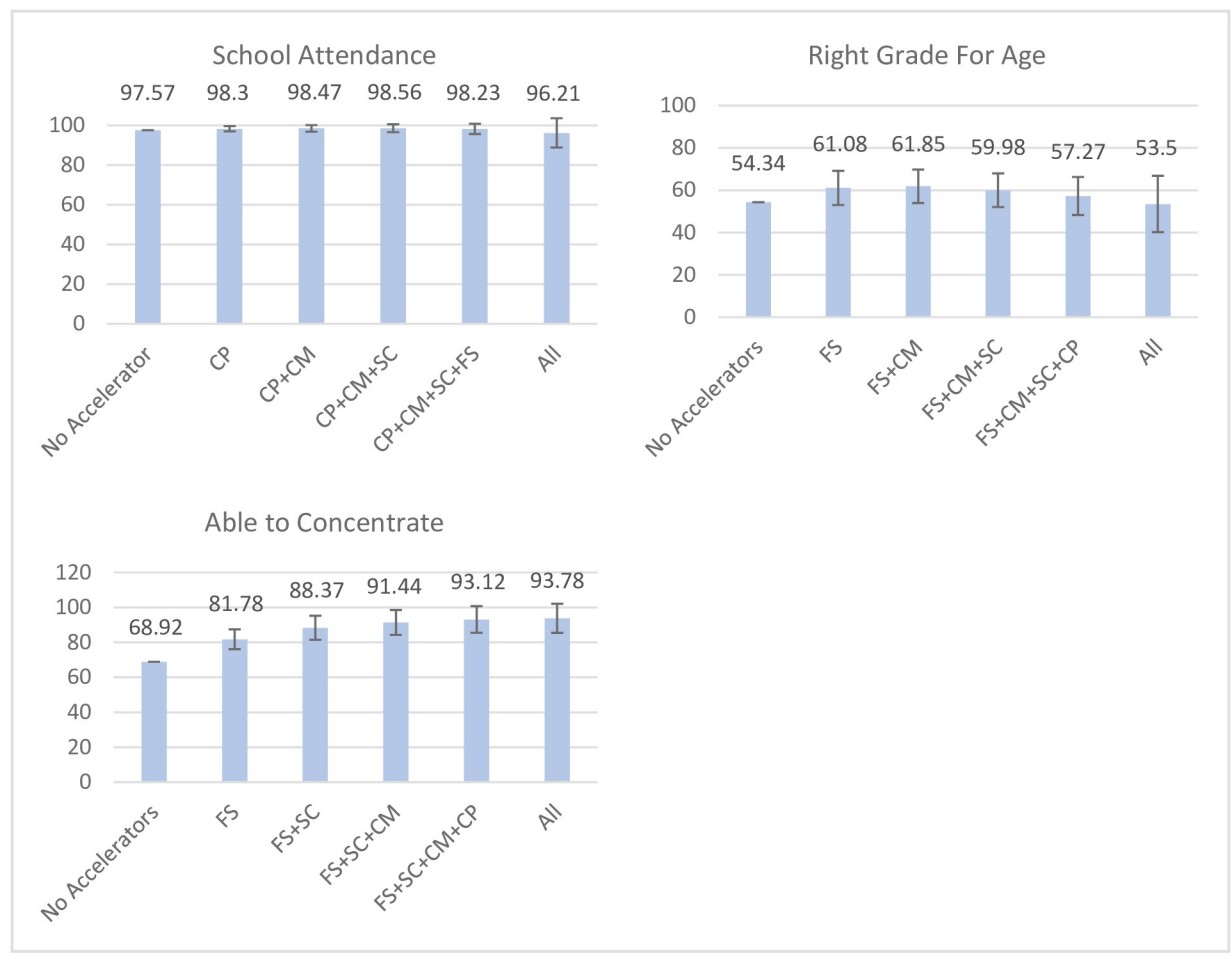

**Fig 3. Accelerator combinations leading to the strongest changes in adjusted probabilities for each school outcome.** Figure scales adjusted based on range of outcome values. CBO = Access to community-based organisations, SC = Living in a Safe Community, CP = Caregiver Praise, FS = Food security, CM = Caregiver Monitoring. 95% confidence intervals included.

praise (-2.79 pp) and living in a safe community (-1.95 pp); however, on its own, food security led to an improvement of + 6.74 pp (rate of being in the right grade for age: 61.08%).

The adjusted probability for <u>being able to concentrate at school</u> lay at 68.92% when no accelerators were present; this rose to 93.78% if all five accelerators were present. Food security led to the largest single-accelerator change (PD: + 12.86 pp), which when additionally combined with living in a safe community led to changes of +19.45 pp, and when caregiver monitoring was subsequently added to a PD of +22.52 pp (ability to concentrate: 91.44%). Only small additional changes resulted from adding food security (PD: +24.20 pp) and having access to all five accelerators (PD: +24.86 pp).

**Violence-related outcomes.** For <u>no violence perpetration</u>, predicted probabilities without any accelerators present lay at 38.49%, which rose to 64.84% with all five accelerators present. Similarly large gains were made through presence of the single accelerators of caregiver praise (PD: +8.71 pp) and CBO access (PD: +8.75 pp), which in combination led to changes of +-17.62 pp. Adding caregiver monitoring led to changes of +23.65 pp (rates of no violence perpetration: 62.14%), with only small additional increases found through adding living in a safe community (+25.74 pp) and being exposed to all accelerators (+26.35 pp) (See Fig 4).

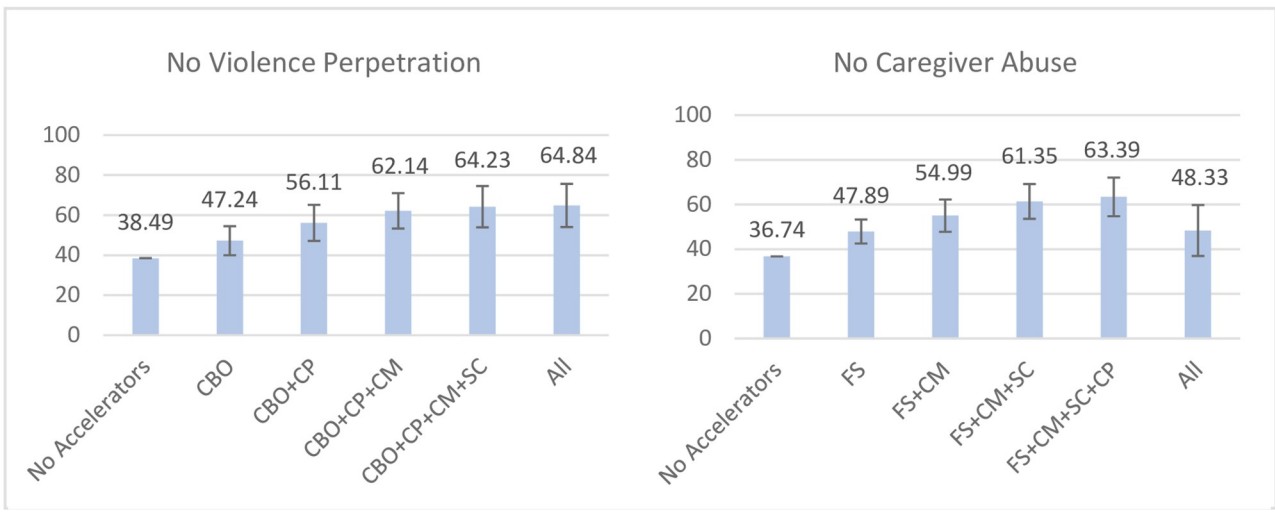

**Fig 4. Accelerator combinations leading to the strongest changes in adjusted probabilities for each violence-related outcome.** Figure scales adjusted based on range of outcome values. CBO = Access to community-based organisations, SC = Living in a Safe Community, CP = Caregiver Praise, FS = Food security, CM = Caregiver Monitoring. 95% confidence intervals included.

Finally, predicted probabilities for no caregiver abuse without accelerators lay at 36.74%. This rose to 63.39% when all accelerators but CBO access were present, with the latter being negatively related to the outcome (PD: -12.89 pp). The largest single accelerator change was achieved through food security (PD: +11.15 pp), which yielded similarly large improvements in combination with living in a safe community (+17.68 pp) and caregiver monitoring (+-18.25 pp). All three together led to a PD of +24.61 pp (no caregiver abuse: 61.35%), with only small additional changes through adding caregiver praise (+26.65 pp).

### Determining optimal numbers of accelerators

Fig 5 depicts an overlay of Figs 1–4 onto the same axis. It illustrates that for most SDG outcomes, the presence of a single or two accelerators seem to lead to the largest changes in predicted probabilities, with the presence of additional accelerators contributing only smaller additive benefits. However, in several cases, various different combinations of accelerators led to similar levels of improvement. For instance, for the SDG-related outcome of no sexual debut, caregiver monitoring (+5.19 pp), living in a safe community (+4.46 pp) and parental praise (+4.23 pp) yielded almost similarly large changes (see S2 Table). Similarly, for no peer problems, combining CBO access with food security (+31.89 pp), caregiver praise (+32.38 pp) or caregiver monitoring (+34.44 pp) all led to substantial gains, indicating that at least for some outcomes, different combinations of accelerators could have similarly large effects.

Table 2 displays predicted probability changes for each SDG outcome under the presence of no accelerator, the most effective single accelerator, the most effective combinations of two, three and four accelerators, and all five accelerators. This highlights to what degree the inclusion of additional accelerators leads to significant improvements in each outcome. Overall, the number of accelerators effective for improving outcomes varied between one and five, depending on the type of outcome and its base rate, and diminishing returns were found as more accelerators were added. More specifically, we found that access to community-based organisations, caregiver praise and food security seemed to be particularly valuable as primary accelerators, with living in a safe community often being in second or third position. Furthermore,

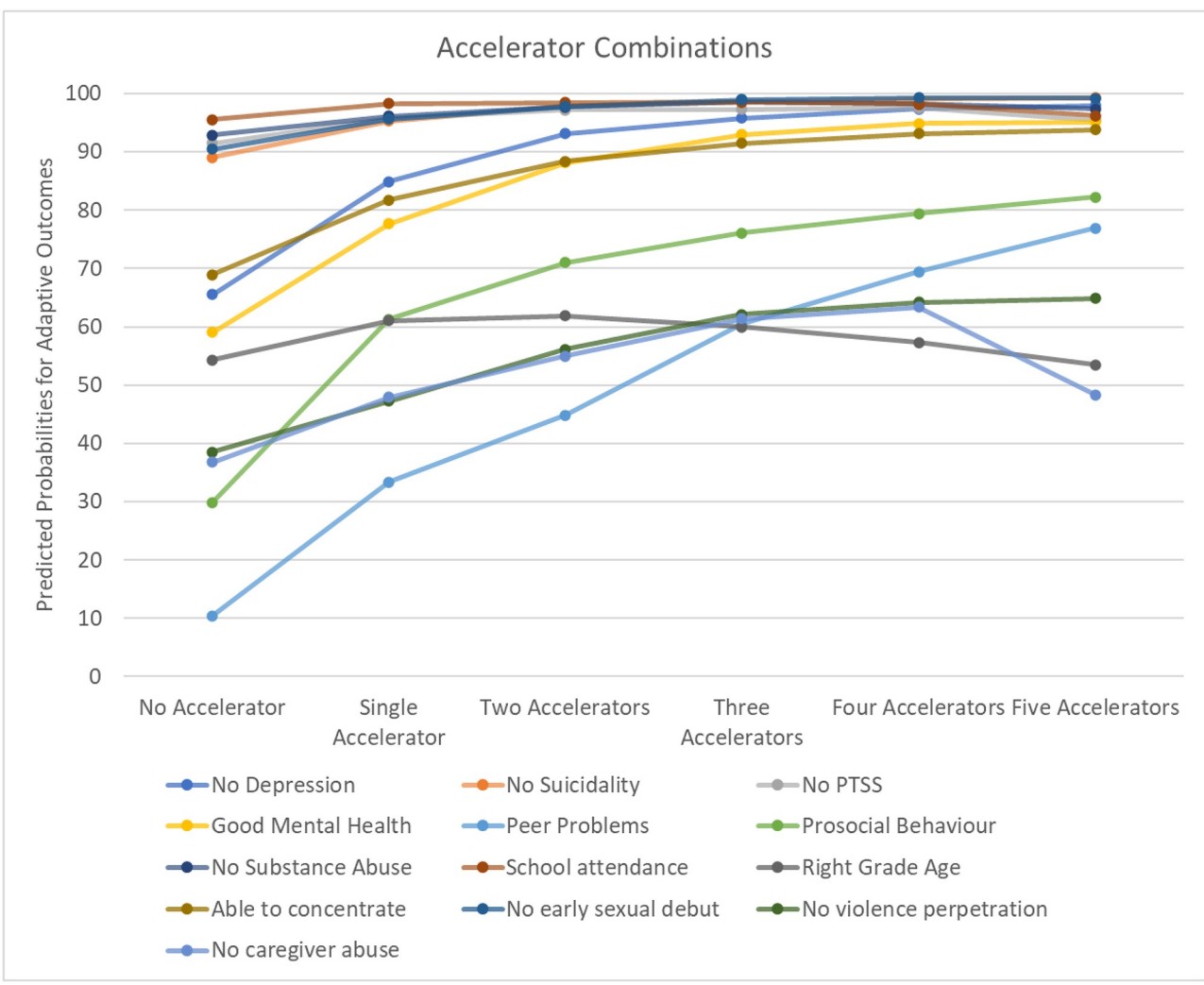

**Fig 5. Accelerator combinations leading to the strongest changes in adjusted probabilities for SDG outcomes.** This figure depicts the predicted probabilities for an adaptive outcome for all 13 indicators, under the presence of no accelerator, the single most effective accelerator, the most effective combinations of two, three and four accelerators, and all five accelerators.

when comparing our combined effects in this paper to what accelerators were associated with specific outcomes in our initial paper [13], we found no evidence of hidden accelerator effects that only became apparent once a certain accelerator was combined with another one. Rather, improvements were only found for combinations of accelerators that had already been found to predict the respective SDGs on their own.

## Discussion

In the current climate of limited resources and increasing needs of children and adolescents in low-and-middle income settings, governments and donors are having to make difficult decisions [26]. It is essential that those decisions are guided by the best possible evidence-base, in order to maximise benefits for young people [27]. Currently, there is a strong research focus on testing new, single interventions. The reality however is that we already know a lot about what protective factors may address various challenges in LMIC. What we do not yet know is

**Table 2. Summary of relative change in outcome probabilities conditional on incremental combinations of protective factors.**

| No Depression | Probability Difference (%) | 95% Confidence Interval |
|---|---|---|
| CBO vs. no accelerator | 19.40 | 11.18; 27.62 |
| CBO+ SC vs. CBO only | 8.22 | 4.23; 12.21 |
| CBO+SC+CP vs. CBO+SC | 2.66 | .66; 4.67 |
| CBO+SC+CP+FS vs. CBO+SC+CP | 1.60 | .14; 3.05 |
| All accelerators vs. CBO+AC+CP+FS | .56 | -.37; 1.49 |
| **No Suicidality** | | |
| CP vs. no accelerator | 6.24 | 2.12; 10.36 |
| CP+CBO vs. CP only | 2.63 | .07; 5.19 |
| CP+CBO+FS vs. CP+CBO | .72 | -.45; 1.90 |
| CP+CBP+FS+SC vs. CP+CBO+FS | .48 | -.21; 1.17 |
| All accelerators vs. CP+CBP+FS+SC | .21 | -.34; .76 |
| **No PTSD** | | |
| CP vs. no accelerator | 4.30 | 1.51; 7.10 |
| CP+SC vs. CP only | 1.41 | -.16; 2.99 |
| CP+SC+FS vs. CP+SC | .26 | -.99; 1.51 |
| CP+SC+FS+CBO vs. CP+SC+FS | .11 | -1.39; 1.60 |
| All accelerators vs. CP+SC+FS+CBO | -2.07 | -3.78; -.36 |
| **Good Mental Health** | | |
| CBO vs. no accelerator | 18.59 | 10.32; 26.87 |
| CBO+SC vs. CBO | 10.48 | 6.34; 14.62 |
| CBO+SC+CP vs. CBO+SC | 4.81 | 2.04; 7.59 |
| CBO+SC+CP+FS vs. CBO+SC+CP | 1.92 | .03; 3.81 |
| All accelerators vs. CBO+SC+CP+FS | .22 | -1.21; 1.67 |
| **No Peer Problems** | | |
| CBO vs. no accelerator | 22.94 | 16.02; 29.85 |
| CBO+CM vs. CBO | 11.51 | 5.99; 17.02 |
| CBO+CM+SC vs. CBO+CM | 15.60 | 9.73; 21.47 |
| CBO+CM+SC+CP vs. CBO+CM+SC | 9.03 | 3.90; 14.16 |
| All accelerators vs. CBO+CM+SC+CP | 7.47 | 1.77; 13.17 |
| **Prosocial Behaviour** | | |
| CBO vs. no accelerator | 31.46 | 23.98; 38.94 |
| CBO+CP vs. CBO | 9.74 | 5.22; 14.27 |
| CBO+CP+SC vs. CBO+CP | 5.08 | 1.03; 9.13 |
| CBO+CP+SC+CM vs. CBO+CP+SC | 3.31 | -.69; 7.31 |
| All accelerators vs. CBO+CP+SC+CM | 2.84 | -.93; 6.62 |
| **No Substance Abuse** | | |
| CP vs. no accelerator | 3.28 | .80; 5.75 |
| CP+CBO vs. CP | 1.58 | -.29; 3.46 |
| CP+CBO+SC vs. CP+CBO | .90 | -.05; 1.86 |
| CP+CBO+SC+CM vs. CP+CBO+SC | -.39 | -1.17; .39 |
| All vs. CP+CBO+SC+CM | -.01 | -.02; .00 |
| **No Early Sexual Debut** | | |
| CM vs. no accelerator | 5.19 | 1.58; 8.79 |
| CM+SC vs. CM | 2.15 | -.06; 4.35 |
| CM+SC+CP vs. CM+SC | 1.06 | -.03; 2.15 |
| CM+SC+CP+CBO vs. CM+SC+CP | .44 | -.49; 1.36 |

*(Continued)*

**Table 2.** (Continued)

| No Depression | Probability Difference (%) | 95% Confidence Interval |
|---|---|---|
| All accelerators vs. CM+SC+CP+CBO | -.10 | -.48; .29 |
| **School Attendance** | | |
| CP vs. no accelerator | .73 | -.60; 2.06 |
| CP+CM vs. CP | .18 | -.86; 1.22 |
| CP+CM+SC vs. CP+CM | .08 | -.88; 1.05 |
| CP+CM+SC+FS vs. CP+CM+SC | -.33 | -1.46; .81 |
| All accelerators vs. CP+CM+SC+FS | -2.02 | -4.97; .92 |
| **Right Grade Age** | | |
| FS vs. no accelerator | 6.74 | 1.32; 12.17 |
| FS+CM vs. FS | .77 | -4.38; 5.91 |
| FS+CM+SC vs. FS+CM | -1.87 | -6.74; 3.01 |
| FS+CM+SC+CP vs. FS+CM+SC | -2.72 | -7.44; 2.01 |
| All accelerators vs. FS+CM+SC+CP | -3.76 | -11.15; 3.62 |
| **Able to Concentrate** | | |
| FS vs. no accelerator | 12.86 | 7.18; 18.54 |
| FS+SC vs. FS | 6.59 | 2.75; 10.43 |
| FS+SC+CM vs. FS+SC | 3.07 | .09; 6.06 |
| FS+SC+CM+CP vs. FS+SC+CM | 1.67 | -.42; 3.76 |
| All accelerators vs. FS+SC+CM+CP | .67 | -2.07; 3.40 |
| **No violence perpetration** | | |
| CBO vs. no accelerator | 8.75 | 1.50; 16.00 |
| CBO+CP vs. CBO | 8.88 | 4.10; 13.65 |
| CBO+CP+CM vs. CBO+CP | 6.02 | .97; 11.18 |
| CBO+CP+CM+SC vs. CBO+CP+CM | 2.10 | -2.61; 6.81 |
| All accelerators vs. CBO+CP+CM+SC | .61 | -4.48; 5.69 |
| **No caregiver abuse** | | |
| FS vs. no accelerator | 11.15 | 5.76; 16.53 |
| FS+CM vs. FS | 7.10 | 1.68; 12.51 |
| FS+CM+SC vs. FS+CM | 6.37 | 1.34; 11.40 |
| FS+CM+SC+CP vs. FS+CM+SC | 2.04 | -2.58; 6.66 |
| All accelerators vs. FS+CM+SC+CP | -15.06 | -22.61; -7.50 |

Notes. For each outcome, we first calculated improvements in predicted outcome probabilities for the single most effective accelerator, as compared to no accelerator being present. Then, improvements were calculated for the presence of the most effective combination of two accelerators, as compared to the single most effect accelerator. Similar steps were performed for the three and four most effective, as well as five accelerators. CBO = Access to community-based organisations, SC = Safe Community, CP = Caregiver praise, FS = Food Security, CM = Caregiver monitoring.

how to combine a number of different factors for an additive benefit to accelerate progress to the best possible effect. The current study provides some initial evidence from a South African context.

Our findings highlight that multiple protective factors may combine to affect SDG outcomes in a step-wise additive way, with stacking effects. For measures of no depression, for instance, the protective factor CBO services was associated with the highest single improvement (65.5% to 83.9%). The combination of CBO access plus living in a safe community, and additional three-fold combination of adding parental praise were each associated with even

higher probability of no depression, 93.1% and 97.9% respectively. Analyses indicate that additional accelerators are only associated with relatively small increases in the probability of no depression. Similar patterns were found across a range of other outcomes, though findings may differ when there are extensive burdens or high base rates of SDG outcomes. As an example of the former, the probability of "no peer problems" without accelerators present lay at 10.4%. Any single protective factor was associated with an increase in the probability of this outcome, at 14.6% in the presence of food security, and 33.3% in the presence of CBO access. Two protective factors, with a variety of combinations, had notable impact, though the highest probability of study outcomes was consistently achieved with all hypothesized protective factors present (76.9%). On the other hand, for outcomes with high base rates (e.g., suicidality, early sexual debut), single protective factors often made the largest difference, with combinatory approaches only leading to small additional increases, though ultimately predicted outcome rates of almost 100%. Such patterns should be considered when planning interventions.

Our data clearly show that different outcomes may require different combinations of protective factors, which is in accordance with what has been found in previous accelerator studies [e.g., 8,11,17,28]. The pattern emerging from the current data seems to suggest that two to three protective factors can often provide a substantial boost, while additional factors provide for marginal gain and diminishing returns. Our data also highlight that some protective factors may have equal impact for the same outcome, and choice decisions can be enhanced by understanding optimum and interchangeable combinations. These findings suggest however that combination approaches do not need to be overly complex, and that good impact can be achieved with two to three combinations.

As found in previous studies [11,12,14], some protective factors were cross-cutting and have traction with multiple outcomes. CBO access and community safety, as well as CBO access plus good parenting often appeared in a cluster associated with a large change in the probability of outcomes. Food security appeared to be particularly important for some school-related outcomes. It may now be necessary to understand components of accelerators so that interventions can be clustered into a typology. For the current study, there appear to be three different typologies—meeting of needs (such as food, cash); care provision (parenting, praise, monitoring), and environmental provision (safe communities, CBO access). Further research would be needed to see if such typology provides insight into efficacy and whether selections should be from one level or across different levels to maximise impact. Our data also shows that some outcomes (e.g., school enrolment and attendance) are more difficult to influence, and different interventions may need to be trialled (e.g., safe schools) [8]. Future research should look to extend these findings, by considering additional accelerators and modelling their cost-effectiveness alone and in combination.

In accordance with the UNDP SDG Accelerator and Bottleneck assessment, a next step will be to investigate drivers of access to the identified accelerators, and interventions that can alter access levels [10]. The current results suggest that cost-effectiveness analyses could be key for implementing sufficient but not excessive provisions in order to maximise gain and support the best value for money [29,30]. When there are constraints such as financial resources or intervention expertise, optimum combinations, or strategic alternatives can be planned. If policy makers aim to impact a wider range of SDGs than covered in the current study, a larger portfolio of interventions may be required [31,32]. The current data provide a first guidance on what combinations of protective factors may be effective. However, further research will be needed to establish a broad evidence base of combinatory approaches, and potential interactive/synergy effects between provisions need to be studied (e.g., interactive effects found for food security and cash transfers) [7]. It is also important to note that the current findings stem from a South African context. While similarities to other SSA countries and LMIC exists, there

may also be important differences (e.g., in health, school and social support systems). Thus, our findings require replication across different countries and cultures.

The current study is a detailed attempt to unpack and understand accelerators and their combinations. However, our results should be interpreted with some caution. The study was not set up directly to investigate accelerators, and our observations should be replicated in more rigorous research designed specifically for this purpose. Furthermore, while the number of protective factors investigated in our original study was relatively extensive, there may be other important accelerators that were not assessed and that should be covered in future studies. Third, the findings should be interpreted within the context of some limitations. For some of the measures, the use of cut offs may have restricted generalizability. For some of the analyses, there were small cell sizes, which affected power and reliability. Furthermore, we performed secondary analyses of observational data, which does not allow for investigations of causality. Future studies may also aim to investigate the effects of individual accelerators while other accelerators are held at their population means instead of zero, to increase external validity of the findings. The current study however focused on isolating effects of protective factors and their combinations. Finally, the age range of the participants may also have an impact on various outcomes and these analyses may not generalise to older (or younger) age groups, and replications across other countries and contexts are required. However, the strengths of the study are that the data are based on real world field samples with high recruitment and follow up rates, testing real world protective factors.

In conclusion, the current study moves the understanding of accelerators a step forward. We suggest that some accelerators resonate across outcomes and may be strong candidates for country support (such as community-based organisation support, parenting, safe communities). If resources are constrained, a maximum of three combinations may provide notable traction, and the additional cost of further interventions may need justification.

## Supporting information

**S1 Table. Access rates to different accelerator combinations (% of overall population) in the study sample.** This table captures which accelerators an individual had access to. E.g., if a person is grouped under "CBO+FS", this means they did not have access to CM, SC, and CP. (DOCX)

**S2 Table. Adjusted probabilities, probability ratios and probability differences for each outcome, given the presence of no, single or different accelerators in combination.** (DOCX)

## Acknowledgments

Acknowledgments to our funders, as well as Prof. Mark Orkin, Dr. Elona Toska, Dr. Heidi Stoeckl and Dr. Lucas Hertzog for their feedback on the current analyses. We would also like to thank Zena Jacobs from Stellenbosch University, as well as the community-based organisations, data collectors and families who participated in the study.

## Author Contributions

**Conceptualization:** Lorraine Sherr, Mark Tomlinson, Sarah Skeen, Franziska Meinck, Chris Desmond, Lucie Cluver.

**Data curation:** Kathryn J. Steventon Roberts.

**Formal analysis:** Katharina Haag.

**Funding acquisition:** Lorraine Sherr, Mark Tomlinson, Lucie Cluver.

**Investigation:** Sarah Skeen, Franziska Meinck, Stefani M. Du Toit, Sarah L. Gordon.

**Methodology:** Lorraine Sherr, Katharina Haag, William E. Rudgard, Chris Desmond, Lucie Cluver.

**Project administration:** Sarah Skeen, Kathryn J. Steventon Roberts.

**Resources:** Sarah Skeen, Franziska Meinck.

**Software:** Katharina Haag, William E. Rudgard.

**Supervision:** Lorraine Sherr, Mark Tomlinson.

**Visualization:** Katharina Haag.

**Writing – original draft:** Lorraine Sherr, Katharina Haag.

**Writing – review & editing:** Mark Tomlinson, William E. Rudgard, Sarah Skeen, Franziska Meinck, Stefani M. Du Toit, Kathryn J. Steventon Roberts, Sarah L. Gordon, Chris Desmond, Lucie Cluver.

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
