## [Decision Letter · Decision Letter 0]

25 May 2022

PONE-D-21-33618Understanding accelerators to improve outcomes for adolescents – an investigation into the nature and quantum of additive effects of protective factors to guide policy making

PLOS ONE

Dear Dr. Haag,

Thank you for submitting your manuscript to PLOS ONE. Firstly, we would like to apologize for the delay in processing your manuscript. It has been exceptionally difficult to secure reviewers to evaluate your study. We have now received two completed reviews, which are available below.

After careful consideration, we feel that it has merit but does not fully meet PLOS ONE’s publication criteria as it currently stands. Therefore, we invite you to submit a revised version of the manuscript that addresses the points raised during the review process. In particular, Reviewer #1 has raised significant scientific concerns about the study that need to be addressed.

 Please submit your revised manuscript by Jul 08 2022 11:59PM. If you will need more time than this to complete your revisions, please reply to this message or contact the journal office at plosone@plos.org. Please include the following items when submitting your revised manuscript:A rebuttal letter that responds to each point raised by the academic editor and reviewer(s). You should upload this letter as a separate file labeled 'Response to Reviewers'.A marked-up copy of your manuscript that highlights changes made to the original version. You should upload this as a separate file labeled 'Revised Manuscript with Track Changes'.An unmarked version of your revised paper without tracked changes. You should upload this as a separate file labeled 'Manuscript'.

We look forward to receiving your revised manuscript.

Kind regards,

Miquel Vall-llosera Camps

Senior Editor

PLOS ONE

Journal Requirements:

3. We noted in your submission details that a portion of your manuscript may have been presented or published elsewhere. [As mentioned in the letter to the editor, a paper presenting the initial statistical analyses leading to the identification of the 5 accelerators covered in the current paper is presently under review at World Development.

The current paper moves substantially beyond this, aiming to identify additive effects and optimal combinations of accelerators, with a stronger focus on policy guidance.

] Please clarify whether this [conference proceeding or publication] was peer-reviewed and formally published. If this work was previously peer-reviewed and published, in the cover letter please provide the reason that this work does not constitute dual publication and should be included in the current manuscript.

Reviewers' comments:

Reviewer's Responses to Questions

**Comments to the Author**

1. Is the manuscript technically sound, and do the data support the conclusions?

Reviewer #1: No

Reviewer #2: Partly

2. Has the statistical analysis been performed appropriately and rigorously? 

Reviewer #1: No

Reviewer #2: I Don't Know

3. Have the authors made all data underlying the findings in their manuscript fully available?

Reviewer #1: No

Reviewer #2: Yes

4. Is the manuscript presented in an intelligible fashion and written in standard English?

Reviewer #1: Yes

Reviewer #2: Yes

5. Review Comments to the Author

Reviewer #1: Major issues:

1. I am not sure if the word of “accelerators” is a widely and academically accepted term.

2. P4 “Results show that various accelerator combinations are effective, though different combinations are needed for different outcomes. Some accelerators ramified across multiple outcomes. An overall analysis showed that the presence of up to three accelerators was associated with marked improvements over multiple outcomes.”: could it possible to make the abstract more quantitative than descriptive?

3. P7-8 “The current analyses build on a previous manuscript (13)” “can be found in the original paper (13).” P10 “Data in our initial paper (Haag et al., under review) furthermore indicated that”: Ref 13 was “under review” so please add reference once it is accepted so that readers will have access to this important reference. If not yet accepted, the readers must upload it to an open access preprint website. Otherwise, the readers miss a key reference that this paper builds upon.

4. P11 “As described above, only five of the seven protective factors originally investigated were related to three or more SDG outcomes and thus defined as accelerators.”: where can I see that 5 out of 7 factors were related to three or more SDGs from Table 1? I do not get it. Why the other 2 factors were not defined as accelerators? At least, from data presented in this paper, I could not make this conclusion.

5. I am not sure if it is my own problem but I do have big difficulty in interpreting Table 1 and S1:

A. E.g. in Table S1, number and percentage of “Food Security (FS)” is 121 and 70%; but in Table 1, number and percentage of “Food Security” is 1446 and 83.1% for baseline, 1438 and 83.0% for retention. Why numbers in Table S1 and Table 1 are so different?

B. Table S1 puzzles me for the combinations: if CBO Access is 10 (0.6%), Food Security is 121 (7.0%), how can their combination be 45 (2.6%), which is higher than CBO? Combination access should refers to the intersection subgroup, right?

6. How are all the probabilities calculated in Table S2? Probability should be different for every individual since their variables (access to accelerators, covariates etc.) are different. What do these adjusted probabilities mean as they are likely for the study sample?

7. Figures 1-4: bar plots always start at 0.

Ref: http://www.chadskelton.com/2018/06/bar-charts-should-always-start-at-zero.html

The choice of start points at 50, 60, or even 85 in Fig 1-4 is very misleading for readers to mistakenly “zoom” the “bigger than appearing” differences.

8. P17 “When investigating the primary accelerators that were associated with each outcome (Table 2), we found that access to community-based organisations, caregiver praise and food security seemed to be particularly valuable as primary accelerators, with living in a safe community often being in second or third position.”: Table 2 lacks any quantitative metric to support the listing of the selected factors for each outcome. In addition, what does the “first”, “second”, and “third” mean? How are the factors ordered?

Minor issues:

1. P4 “This data has also clearly shown” and “the current data shows the detailed impact”: I may be wrong since I am not a native English speaker, but is the word “data” of plural form so should use “have” instead of “has”? Actually the authors did have “Data were collected at baseline”.

2. P4 “Measures in common between the two databases were used to generate five accelerators (caregiver praise, caregiver monitoring, food security, living in a safe community, and access to community-based organizations and to investigate”: missing “)” after “organizations”.

3. P5 “In order to meet the UNDP Sustainable Developmental Goals (SDGs)”: UNDP should be annotated as not every author knows this acronym.

4. P7 “For the CCC study, Ethical approval”: “ethical”.

5. P7 “to investigate the effects of seven hypothesized protective factor on 14 SDG-related outcomes.”: “factors”.

6. P8 “exposed to community violence; 4) consistent”: “;” to “,”.

7. P8 “relatively young) (SDG 5.6); no delinquent behaviours; and no exposure to physical and emotional abuse by the caregiver (SDGs 16.1 and 16.2).”: which SDG does “no delinquent behaviours” belong to?

8. P8 “Exceptions were the SDQ subscales”: what does SDQ stand for?

9. P9 “This group was on average younger at baseline (M = 11.23, SD = 1.25, versus M =11.47, SD = 1.19, p = .046)”: 1) “SD” to “SD”; 2) “p” to “P”; 3) I highly recommend changing “.046” to “0.046” for all such numeric expressions in this paper.

10. P10 “Any grant (T2 only)”: what is T2?

11. P11: format spaces accordingly.

12. P11 “drop in numbers may be due to changes in caregiver between BL and FU (n = 401, 23.0%)”: what is BL and what is FU?

13. P12 “The adjusted probability of experiencing no depression if no accelerator” and “The adjusted probability for no suicidal ideation without any accelerators”: why are there underlines?

14. P12 “The adjusted probability of experiencing no depression if no accelerator was present lay at 65.50%, which was raised to 97.94% if all five accelerators were present”: generally speaking, when taking about probability, we use 0-1 as the range and commonly use the probability is 0.655 rather than 65.50%.

15. Figure 5 lines overlap a lot. Please make the lines partially transparent (set alpha value) to improve readability.

Reviewer #2: Dear Authors,

I read the manuscript with a great interest, both contents-wise but even more eager to see the statistical methodology used. To identify optimum combinations of accelerators is extremely needed nowadays.

Few proposals:

1. The title is not specific enough - reader can not really understand what topic is the paper dealing with. I suggest to add "SDG-related" to the title: "Understanding accelerators to improve SDG-related outcomes for adolescents...".

2. Abstract: Methods of statistic analysis are missing.

3. It is difficult to understand fully the methodology of the paper without reading also the linked-paper from Supplement 3.

4. Materials and methods / Analysis. Two thirds of quite modest paragraph are describing methodology of data analysis already published in the previous (linked) paper. The in-depth analysis, presented in current paper, is insufficiently presented. The model used to calculate adjusted probabilities and adjusted probability differences for all possible combinations of accelerators should be described in detail in the present paper, to make possible for the reader to understand fully the data analysis performed.

5. Line 334: AD - is this correct?

6. Figures 1 - 4: I would prefer to keep the scales of the Y-axes in the same range.

7. It is most welcome to see among authors also native researchers from the country of research origin.

Congratulations!

6. PLOS authors have the option to publish the peer review history of their article (what does this mean?). If published, this will include your full peer review and any attached files.

Reviewer #1: No

Reviewer #2: **Yes: **Pia Vracko

---

## [Author Response · Author response to Decision Letter 0]

19 Sep 2022

Review Comments to the Author

Reviewer #1: 

We would like to thank reviewer 1 for their helpful and detailed feedback, which has improved the quality and clarity of our submission substantially.

Major issues:

1. I am not sure if the word of “accelerators” is a widely and academically accepted term.

The term “accelerators” was coined as a key concept by the UNDP in 2017. Accelerators are defined as protective factors that positively affect multiple SDG-related outcomes, access to which can be altered through intervention. An initial paper by Cluver and colleagues in 2019 aimed to quantitively operationalize this term. Since then, an increasing amount of research has been published on potential accelerators in the Sub-Saharan African context (e.g., Cluver et al., 2020, Du Toit et al., 2022, Mehbrahtu et al., 2021, Meinck et al., 2021) and related research approaches are continuously being developed and refined. Thus, we believe that “accelerators” is the correct term to use in our paper in order to link it to the emerging literature in the field.

2. P4 “Results show that various accelerator combinations are effective, though different combinations are needed for different outcomes. Some accelerators ramified across multiple outcomes. An overall analysis showed that the presence of up to three accelerators was associated with marked improvements over multiple outcomes.”: could it possible to make the abstract more quantitative than descriptive?

Since the type and number of relevant accelerators vary for each of the 13 outcomes, it was not possible for us to sum this up using quantitative data in a way succinctly enough to remain within the word limits of the abstract. 

3. P7-8 “The current analyses build on a previous manuscript (13)” “can be found in the original paper (13).” P10 “Data in our initial paper (Haag et al., under review) furthermore indicated that”: Ref 13 was “under review” so please add reference once it is accepted so that readers will have access to this important reference. If not yet accepted, the readers must upload it to an open access preprint website. Otherwise, the readers miss a key reference that this paper builds upon.

This manuscript has in the meantime been accepted for publication in World Development. PlosOne was notified of this via an email on 03.11.2021. The manuscript is now available at https://www.sciencedirect.com/science/article/pii/S0305750X21003545?via%3Dihub

4. P11 “As described above, only five of the seven protective factors originally investigated were related to three or more SDG outcomes and thus defined as accelerators.”: where can I see that 5 out of 7 factors were related to three or more SDGs from Table 1? I do not get it. Why the other 2 factors were not defined as accelerators? At least, from data presented in this paper, I could not make this conclusion.

We have aimed to make it clearer in the text that only five out of the seven potential protective factors in our original analyses met our definition of being an accelerator, i.e. were related to at least three or more SDG outcomes. A reference to the original paper and a note that this paper contains the detailed base analyses have now also been added.

5. I am not sure if it is my own problem but I do have big difficulty in interpreting Table 1 and S1:

A. E.g. in Table S1, number and percentage of “Food Security (FS)” is 121 and 70%; but in Table 1, number and percentage of “Food Security” is 1446 and 83.1% for baseline, 1438 and 83.0% for retention. Why numbers in Table S1 and Table 1 are so different?

Table 1 shows how many individuals overall were food secure at baseline and follow-up respectively, in this case 83% of the retained sample. Of these individuals, some may have had access to several other accelerators, and some may have only been food secure. This is captured in Table S1, which provides information on how many individuals had access to a particular combination of accelerators out of the whole sample. The 7% remarked upon by the reviewer are individuals who were only food secure but did not have access to any other accelerators. Thus, the two tables address somewhat different questions, with Table S1 potentially being of particular interest to policy makers, as it reflects how common the different accelerator combinations were in our sample.

We have aimed to make it clearer in the main body of the text how the two tables are different and have provided Table S1 with a different heading and a legend describing it in more detail.

B. Table S1 puzzles me for the combinations: if CBO Access is 10 (0.6%), Food Security is 121 (7.0%), how can their combination be 45 (2.6%), which is higher than CBO? Combination access should refers to the intersection subgroup, right?

As described above, table S1 allocates each individual to exactly the one combination of accelerators they had access to. So the label “food security" means that this individual was only food secure, but did not have access to any other accelerators, while the group that had “CBO access AND food security“ captures a different group of individuals. Table labels and the table legend have been updated to reflect this more clearly.

6. How are all the probabilities calculated in Table S2? Probability should be different for every individual since their variables (access to accelerators, covariates etc.) are different. What do these adjusted probabilities mean as they are likely for the study sample?

The probabilities are estimated from our overall model using marginal effects. They reflect how likely it is that each SDG-related outcome was met under each accelerator combination, e.g. “no depression” under “access to food security but no other accelerator”, keeping the covariates constant. As described above, Table S1 gives an indication of what proportions of individuals in our sample had access to each accelerator combination. We hope that the combination of this information can provide valuable insights to researchers and policy makers.

7. Figures 1-4: bar plots always start at 0.

Ref: http://www.chadskelton.com/2018/06/bar-charts-should-always-start-at-zero.html

The choice of start points at 50, 60, or even 85 in Fig 1-4 is very misleading for readers to mistakenly “zoom” the “bigger than appearing” differences.

We agree and have changed all graphs to start at zero as suggested by the reviewer.

8. P17 “When investigating the primary accelerators that were associated with each outcome (Table 2), we found that access to community-based organisations, caregiver praise and food security seemed to be particularly valuable as primary accelerators, with living in a safe community often being in second or third position.”: Table 2 lacks any quantitative metric to support the listing of the selected factors for each outcome. In addition, what does the “first”, “second”, and “third” mean? How are the factors ordered?

To provide stronger quantitative analyses supporting our conclusions, we have now calculated probability differences between the presence of none, and the most effective combinations of one, two, three and four, as well as all five accelerators for each outcome, including confidence intervals. This highlights more clearly to what degree the inclusion of additional accelerators led to a significant improvement in each outcome. We have now changed Table 2 to display these calculations.

Minor issues:

1. P4 “This data has also clearly shown” and “the current data shows the detailed impact”: I may be wrong since I am not a native English speaker, but is the word “data” of plural form so should use “have” instead of “has”? Actually the authors did have “Data were collected at baseline”.

“Data” can be used in both plural and singular form, but we have now adjusted it to the plural form throughout the manuscript to improve consistency.

2. P4 “Measures in common between the two databases were used to generate five accelerators (caregiver praise, caregiver monitoring, food security, living in a safe community, and access to community-based organizations and to investigate”: missing “)” after “organizations”.

This has been added.

3. P5 “In order to meet the UNDP Sustainable Developmental Goals (SDGs)”: UNDP should be annotated as not every author knows this acronym.

The full name has now been written out.

4. P7 “For the CCC study, Ethical approval”: “ethical”.

This has been changed.

5. P7 “to investigate the effects of seven hypothesized protective factor on 14 SDG-related outcomes.”: “factors”.

This has been updated.

6. P8 “exposed to community violence; 4) consistent”: “;” to “,”.

This has been updated.

7. P8 “relatively young) (SDG 5.6); no delinquent behaviours; and no exposure to physical and emotional abuse by the caregiver (SDGs 16.1 and 16.2).”: which SDG does “no delinquent behaviours” belong to?

Delinquent behaviors in adolescents often comprise aggressive and/or violent behaviors. Thus, we have grouped them under SDG 16.1, “Reduce violence everywhere”. This is now mentioned in the manuscript.

8. P8 “Exceptions were the SDQ subscales”: what does SDQ stand for?

This has now been replaced by terms capturing the content of the respective scales (prosocial behavior, peer problems)- information on the underlying measure (SDQ= Strength and Difficulties Questionnaire) is available in the base paper, and a reference clarifying this is provided at the beginning of the methods section.

9. P9 “This group was on average younger at baseline (M = 11.23, SD = 1.25, versus M =11.47, SD = 1.19, p = .046)”: 1) “SD” to “SD”; 2) “p” to “P”; 3) I highly recommend changing “.046” to “0.046” for all such numeric expressions in this paper.

SD has been adjusted to be fully in italics. The use of lowercase p and .046 are common conventions in our field (see APA guidelines). We were not able to find any PlosOne guidelines suggesting the use of one convention over the other but are happy to be advised by the editorial staff and make any necessary changes.

10. P10 “Any grant (T2 only)”: what is T2?

This has been changed to “FU” to reflect “follow-up”, in accordance with the term that has been used throughout the rest of the manuscript

11. P11: format spaces accordingly.

It is not clear from the comment where this applies- we would be happy to make any necessary changes if given further guidance.

12. P11 “drop in numbers may be due to changes in caregiver between BL and FU (n = 401, 23.0%)”: what is BL and what is FU? 

This has been added, BL= baseline, FU = follow-up.

13. P12 “The adjusted probability of experiencing no depression if no accelerator” and “The adjusted probability for no suicidal ideation without any accelerators”: why are there underlines?

The underlines are intended as a visual aid to help to highlight to the reader which SDG we are currently referring to in the text.

14. P12 “The adjusted probability of experiencing no depression if no accelerator was present lay at 65.50%, which was raised to 97.94% if all five accelerators were present”: generally speaking, when taking about probability, we use 0-1 as the range and commonly use the probability is 0.655 rather than 65.50%.

Since percent and probabilities are ultimately equivalent (e.g., 50 per 100 versus 0.5), we have chosen to retain the percent values for ease of interpretation by the reader.

15. Figure 5 lines overlap a lot. Please make the lines partially transparent (set alpha value) to improve readability.

This has been changed.

Reviewer #2: 

Dear Authors,

I read the manuscript with a great interest, both contents-wise but even more eager to see the statistical methodology used. To identify optimum combinations of accelerators is extremely needed nowadays.

We would like to thank reviewer 2 for the positive comments, which we have aimed to address in the following.

Few proposals:

1. The title is not specific enough - reader can not really understand what topic is the paper dealing with. I suggest to add "SDG-related" to the title: "Understanding accelerators to improve SDG-related outcomes for adolescents...".

We agree, and this has been updated.

2. Abstract: Methods of statistic analysis are missing.

This has now been updated to clarify that we calculated predicted probabilities and probability differences to investigate optimal accelerator combinations for 13 SDG outcomes.

3. It is difficult to understand fully the methodology of the paper without reading also the linked-paper from Supplement 3.

We have added more information on the model used for calculating the predicted probabilities as requested by the reviewer (see also next point). The methods from the published baseline paper have been covered only briefly, in order to allow sufficient space to discuss the additive nature of accelerator effects and additional analyses.

4. Materials and methods / Analysis. Two thirds of quite modest paragraph are describing methodology of data analysis already published in the previous (linked) paper. The in-depth analysis, presented in current paper, is insufficiently presented. The model used to calculate adjusted probabilities and adjusted probability differences for all possible combinations of accelerators should be described in detail in the present paper, to make possible for the reader to understand fully the data analysis performed.

This has been expanded upon. Additional analyses have been added calculating probability differences for the presence of none, and the most effective combinations of one, two, three and four, as well as all five accelerators for each outcome, including confidence intervals. The methodology underlying this has been expanded upon in the methods section. The methodology of the base paper has been covered still, to aid the reader in understanding the underlying model.

5. Line 334: AD - is this correct?

This was an error and has been removed.

6. Figures 1 - 4: I would prefer to keep the scales of the Y-axes in the same range.

This has been updated.

7. It is most welcome to see among authors also native researchers from the country of research origin.

Congratulations!

Journal Requirements:

This has been checked and the formatting has been updated accordingly.

This has been added.

3. We noted in your submission details that a portion of your manuscript may have been presented or published elsewhere. [As mentioned in the letter to the editor, a paper presenting the initial statistical analyses leading to the identification of the 5 accelerators covered in the current paper is presently under review at World Development.

The current paper moves substantially beyond this, aiming to identify additive effects and optimal combinations of accelerators, with a stronger focus on policy guidance.

] Please clarify whether this [conference proceeding or publication] was peer-reviewed and formally published. If this work was previously peer-reviewed and published, in the cover letter please provide the reason that this work does not constitute dual publication and should be included in the current manuscript.

The base paper was published in Word Development in November 2021, after a peer review process. PlosOne was notified of this by one of the co-authors (Katharina Haag) on 03.11.2021. The analyses presented in the World Development paper identify the 5 accelerators we refer to in the current paper, and it has a stronger methodology focus. The current study explores additive effects, which are not touched upon at all in the base paper. Additive effects have not been studied before in the accelerator literature but present a clear progression of the field. In particular, our paper aims to elucidate how many protective factors may be able to yield optimal improvements, which is highly relevant for policy making. Overall, overlap between the two manuscripts is limited, except for use of the same base methods.

---

## [Decision Letter · Decision Letter 1]

2 Nov 2022

PONE-D-21-33618R1Understanding accelerators to improve SDG-related outcomes for adolescents – an investigation into the nature and quantum of additive effects of protective factors to guide policy makingPLOS ONE

Dear Dr. Haag,

Thank you for submitting your manuscript to PLOS ONE. After careful consideration, we feel that it has merit but does not fully meet PLOS ONE’s publication criteria as it currently stands. Therefore, we invite you to submit a revised version of the manuscript that addresses the points raised during the review process. If you are able to attend to the final comments of Reviewer 2, I would be delighted to recommend this paper to be accepted for publication.

We look forward to receiving your revised manuscript.

Kind regards,

Lindsay Stark

Academic Editor

PLOS ONE

Journal Requirements:

Reviewers' comments:

Reviewer's Responses to Questions

**Comments to the Author**

1. If the authors have adequately addressed your comments raised in a previous round of review and you feel that this manuscript is now acceptable for publication, you may indicate that here to bypass the “Comments to the Author” section, enter your conflict of interest statement in the “Confidential to Editor” section, and submit your "Accept" recommendation.

Reviewer #2: (No Response)

2. Is the manuscript technically sound, and do the data support the conclusions?

Reviewer #2: Yes

3. Has the statistical analysis been performed appropriately and rigorously? 

Reviewer #2: Yes

4. Have the authors made all data underlying the findings in their manuscript fully available?

Reviewer #2: Yes

5. Is the manuscript presented in an intelligible fashion and written in standard English?

Reviewer #2: Yes

6. Review Comments to the Author

Reviewer #2: Dear Authors,

Your responses address the revision comments. In addition, there are few more comments to the revised paper:

1. In the discussion section, you should discuss whether the results from the study in South Africa are generalizable to other regions/countries or are context specific for South Africa / Sub-Saharan Africa? If yes, to which other populations (geographically)? Is it so only for the synergistic interactions between accelerators or also for the precise factors studied and their distinct combinations? Especially, as the introduction starts with “Adolescents living in Sub-Saharan Africa…”, it is suggestive that the results are valid for this whole region even though the study subjects are only from South Africa (Do authors suggest that South Africa is representative for Sub-Saharan Africa?). In the first paragraph of discussion, though, LMIC are mentioned, leading to the conclusion that the results are valid for all LMIC countries….

2. Keywords: Change “Policy” to “policy”.

3. Figure 5: Label for y axis is missing (or title should be changed). What does 100 % in Figure 5 actually mean? Please keep in mind that some readers may first look at the Figure 5 and they should get enough comprehensive information from the figure alone on what it is about.

4. Supporting information: revise titles “S2 Appendix 2” and “S1 Table”, too add some order to the appendices.

Once these comments are addressed, I consider the paper to be ready for publishing.

7. PLOS authors have the option to publish the peer review history of their article (what does this mean?). If published, this will include your full peer review and any attached files.

Reviewer #2: No

---

## [Author Response · Author response to Decision Letter 1]

7 Nov 2022

Reviewer #2: Dear Authors,

Your responses address the revision comments. In addition, there are few more comments to the revised paper.

We would like to thank reviewer 2 for taking the time to re-review our manuscript and providing a valuable comment about generalizability, which we have now addressed in our manuscript as outlined below.

1. In the discussion section, you should discuss whether the results from the study in South Africa are generalizable to other regions/countries or are context specific for South Africa / Sub-Saharan Africa? If yes, to which other populations (geographically)? Is it so only for the synergistic interactions between accelerators or also for the precise factors studied and their distinct combinations? Especially, as the introduction starts with “Adolescents living in Sub-Saharan Africa…”, it is suggestive that the results are valid for this whole region even though the study subjects are only from South Africa (Do authors suggest that South Africa is representative for Sub-Saharan Africa?). In the first paragraph of discussion, though, LMIC are mentioned, leading to the conclusion that the results are valid for all LMIC countries….

We agree with the reviewer that caution against over-interpretation of our results is warranted. We have now made it clearer that the data are from South Africa and predominantly valid for this context. Given that there are both similarities and differences between South Africa and other Sub-Saharan African countries, we highlight that a replication in other countries is needed.

2. Keywords: Change “Policy” to “policy”.

In our version of the document, this is not capitalized. We are not sure why this was different in the document the reviewer received.

3. Figure 5: Label for y axis is missing (or title should be changed). What does 100 % in Figure 5 actually mean? Please keep in mind that some readers may first look at the Figure 5 and they should get enough comprehensive information from the figure alone on what it is about.

A label for the y axis has been added. Descriptive text has been added to the caption, clarifying the contents of the figure.

4. Supporting information: revise titles “S2 Appendix 2” and “S1 Table”, too add some order to the appendices.

In accordance with the labels recommended by the journal, we have named our Appendices S1 Table and S2 Table, which also reflects the table order. We had already updated this within the manuscript, but have now also labelled the supporting documents accordingly. 

References

In accordance with the recommendations made by the editor, we have checked out reference list to ensure it is complete. We have updated two citations (15,28), which had previously been under review, with the publication details.

---

## [Editor Report · Decision Letter 2]

9 Nov 2022

Understanding accelerators to improve SDG-related outcomes for adolescents – an investigation into the nature and quantum of additive effects of protective factors to guide policy making

PONE-D-21-33618R2

Dear Dr. Haag,

We’re pleased to inform you that your manuscript has been judged scientifically suitable for publication and will be formally accepted for publication once it meets all outstanding technical requirements.

Kind regards,

Lindsay Stark

Academic Editor

PLOS ONE

Additional Editor Comments (optional):

Thank you for attending to the final comments. 
---

## [Editor Report · Acceptance letter]

27 Dec 2022

PONE-D-21-33618R2 

Understanding accelerators to improve SDG-related outcomes for adolescents – an investigation into the nature and quantum of additive effects of protective factors to guide policy making 

Dear Dr. Haag:

I'm pleased to inform you that your manuscript has been deemed suitable for publication in PLOS ONE. Congratulations! Your manuscript is now with our production department. 

Kind regards, 

on behalf of

Dr. Lindsay Stark 

Academic Editor

PLOS ONE